# Circulating Tumor Cells as Biomarkers for Renal Cell Carcinoma: Ready for Prime Time?

**DOI:** 10.3390/cancers15010287

**Published:** 2022-12-31

**Authors:** Anabela Couto-Cunha, Carmen Jerónimo, Rui Henrique

**Affiliations:** 1Integrated Master in Medicine, School of Medicine & Biomedical Sciences, University of Porto (ICBAS-UP), Rua Jorge Viterbo Ferreira 228, 4050-313 Porto, Portugal; 2Department of Pathology and Molecular Immunology, School of Medicine & Biomedical Sciences, University of Porto (ICBAS-UP), Rua Jorge Viterbo Ferreira 228, 4050-313 Porto, Portugal; 3Department of Pathology & Cancer Biology & Epigenetics Group—Research Center of IPO Porto (CI-IPOP)/RISE@CI-IPOP (Health Research Network), Portuguese Oncology Institute of Porto (IPO-Porto)/Porto Comprehensive Cancer Centre Raquel Seruca (P.CCC Raquel Seruca), Rua Dr. António Bernardino de Almeida, 4200-072 Porto, Portugal

**Keywords:** circulating tumor cells, renal cell carcinoma, detection, prognosis, methods

## Abstract

**Simple Summary:**

Renal cell carcinoma (RCC) is the most common form of kidney cancer, characterized by silent progression at early stages, heterogeneous behavior and resistance to chemotherapy and radiotherapy. The clinical challenges posed by RCC require the development of a novel class of biomarkers which may better portray the biology of the disease. Herein, we explore the potential of circulating tumor cells (CTCs) as a source of biomarker information in RCC, emphasizing the more recently published findings, highlighting CTCs biology and molecular characterization through existing and emerging techniques for CTC enrichment and detection, exploring their clinical applications in RCC.

**Abstract:**

Renal cell carcinoma (RCC) is among the 15 most common cancers worldwide, with rising incidence. In most cases, this is a silent disease until it reaches advance stages, demanding new effective biomarkers in all domains, from detection to post-therapy monitoring. Circulating tumor cells (CTC) have the potential to provide minimally invasive information to guide assessment of the disease’s aggressiveness and therapeutic strategy, representing a special pool of neoplastic cells which bear metastatic potential. In some tumor models, CTCs’ enumeration has been associated with prognosis, but there is a largely unexplored potential for clinical applicability encompassing screening, diagnosis, early detection of metastases, prognosis, response to therapy and monitoring. Nonetheless, lack of standardization and high cost hinder the translation into clinical practice. Thus, new methods for collection and analysis (genomic, proteomic, transcriptomic, epigenomic and metabolomic) are needed to ascertain the role of CTC as a RCC biomarker. Herein, we provide a critical overview of the most recently published data on the role and clinical potential of CTCs in RCC, addressing their biology and the molecular characterization of this remarkable set of tumor cells. Furthermore, we highlight the existing and emerging techniques for CTC enrichment and detection, exploring clinical applications in RCC. Notwithstanding the notable progress in recent years, the use of CTCs in a routine clinical scenario of RCC patients requires further research and technological development, enabling multimodal analysis to take advantage of the wealth of information they provide.

## 1. Introduction

Renal cell carcinoma (RCC) is one of the 15 most common cancers in the world, ranking 14th according to the World Health Organization (WHO) [1]. Comparing both sexes, RCC is more frequent in men, being the 9th most diagnosed cancer worldwide, whereas, in women, it is the 14th. In 2020 alone, RCC was responsible for about 2.2% of all new cancer diagnoses and 1.8% of cancer deaths. According to 2020 GLOBOCAN statistics, there were more than 431,000 new cases and near 180,000 deaths from RCC, globally. Of these, men had the highest figures in both incidence and mortality, leading to a higher cumulative risk [1]. Unfortunately, trends indicate that the incidence of kidney cancer has increased in the last decades [2,3] and will continue to increase in the future [4], with about 666,000 new cases and approximately 301,000 deaths expected worldwide in 2040, depending on demographic changes [1].

RCC comprises several different entities. This is well illustrated by the latest WHO classification, based on studies revealing molecular aspects of kidney tumors [5]. At the epithelial level, new entities defined by genetic aberrations have emerged, such as SMARCB1-deficient renal medullary carcinoma, TFEB-mutated RCC, ELOC-mutated RCC and ALK-rearranged RCC. At the morphological level, a new entity, the solid and cystic eosinophilic RCC, was introduced [5]. Beyond the molecular differences among the currently recognized subtypes, there is prognostic variability, different therapeutic management as well as variants with hereditary features to be considered. In terms of frequency, the most common subtype is, by far, clear cell RCC (ccRCC), accounting for 60 to 70% of all renal cancers in adulthood, followed by papillary RCC (pRCC) and chromophobe RCC (chrRCC, mostly sporadic, accounting for 5–7% of all RCC). Regarding RCC subtypes with a hereditary character, they have specific features and are rare, being usually associated with other inherited tumor syndromes with autosomal dominant transmission. These syndromes include von Hippel-Lindau, hereditary leiomyomatosis and renal cell carcinoma (HLRCC) and germline SDH mutations. In terms of prognosis, some subtypes are associated with favorable outcome, such as multilocular cystic renal neoplasm of low malignant potential, tubular mucinous and spindle cell RCC, tubulocystic RCC, RCC associated with acquired cystic disease and papillary clear cell RCC. On the other hand, there are very aggressive subtypes, such as medullary RCC and collecting duct RCC [6].

As most cases of RCC do not present with clinical manifestations, at present, the diagnosis is mostly incidental following medical imaging procedures performed for investigating other conditions [4,7]. Nonetheless, although magnetic resonance imaging (MRI) and positron emission tomography/computed tomography (PET/CT) seem to have a superior diagnostic performance for RCC compared to single photon emission computed tomography (SPECT) [8], imaging techniques fail to detect, globally, more than one-third of potential RCC diagnoses [9]. Additionally, approximately 33–50% of cases present with metastatic disease at diagnosis. Even after curative-intent surgical treatment of RCC, metastases develop in 20–40% of patients [8,10].

Thus, considering the diversity and heterogeneity of RCC at the clinical and pathobiological level, there is an urgent need for biomarkers which might enable not only early detection of localized and metastatic RCC, but also provide information about the prognosis and prediction of response to treatment, allowing for a truly personalized approach [11]. Indeed, many studies have been published over the years concerning the discovery of potential RCC biomarkers, making it a hot topic and one of the cornerstones of research in this field [2].

One of the most studied biomarkers in cancer are circulating tumor cells (CTC). They constitute a promising and minimally invasive strategy to obtain information about RCC, considering that they are derived from the tumor cells, are present in the bloodstream and may lead to metastasis formation. CTCs can be collected through liquid biopsy and then analyzed using several techniques, enabling applications that could revolutionize the way that RCC is detected and managed [12]. Indeed, CTCs have attracted increasing attention from researchers during the last few years, as demonstrated by several studies published in other cancer models, including hepatocellular [13], lung [14], colorectal [15], pancreatic [16], breast, bladder and prostate [17] cancers. In RCC, CTCs constitute a major research topic, as attested by the 251 results extracted from a PubMed query comprising the last 5 years (2017–2022), using the terms “circulating tumor cell” and “renal cancer”.

In this review, we aim to provide a critical overview of the most recently published data about the role and clinical potential of CTCs in RCC. Firstly, we will explore CTCs’ biology, highlighting their heterogeneity and clinical significance. Then, we will summarize the reported enrichment and detection techniques and explore new ones that have recently emerged. Finally, we will describe the possible clinical applications of CTC and their usefulness, as well as address future directions in this field.

## 2. Materials and Methods

An extensive bibliographic search was conducted primarily in the PubMed^®^ database, using the following key terms “CTC”, “circulating tumor cells”, “renal cancer”, “RCC”, “metastatic RCC”, “role”, “biomarker” and “liquid biopsy” alone or in different combinations, also secondarily using the terms “detection”, “enrichment”, “clinical applications”, “diagnosis” and “prognosis”.

The main selection criterion was articles published in the past six years since 2016. Only articles in the English language, with full text available and intrinsically related to the review’s theme, containing the terms previously mentioned, by first checking the title and abstract and then secondarily the body of the article, were selected.

## 3. CTCs Biology and Clinical Significance: An Overview

Circulating tumor cells (CTC) are rare cancer cells that are released from the primary tumor site, nodal or systemic metastases to the bloodstream. Thus, CTCs play a key role in the metastatic cascade, eventually becoming the precursors of metastasis in distant sites [18,19]. This complex process (metastatic cascade, Figure 1) is composed by several stages which include: the ability of cancer cells to invade, the intravasation into the vasculature, the resistance to death in the bloodstream, the extravasation into the host organ tissue and adhesion to it to, finally, allow for cell growth and proliferation [20,21].

In the beginning of this process, cancer cells endure a dynamic transformation in their phenotype so they may acquire invasiveness capabilities, which is called epithelial-to-mesenchymal transition (EMT) [22]. EMT consists of the change from an epithelial to a mesenchymal phenotype, of variable extent, by remodeling morphological and molecular features that together will facilitate cell motility and detachment from the primary tumor, angioinvasion and intravasation, as well as cellular survival in the bloodstream [18,22,23]. Interestingly, a recent study on kidney cancer, that included 34 patients, evaluated and compared the expression of epithelial and mesenchymal markers as well as CTC count, indirectly showing, for the first time, the occurrence of EMT in renal cancer cells [24]. In addition, the EMT process has been investigated in several malignancies such as prostate [25], breast [26], gastrointestinal [27] and colorectal [28] cancers, among others. Interestingly, EMT seems to be, in part, regulated by long noncoding RNAs (lncRNAs), as demonstrated by Xia et al., who performed a study on 511 ccRCC patients and developed a panel of 12 differentially expressed EMT-related lncRNA that may serve as a prognostic signature [29].

One of the most important steps in EMT phenotype change is cytoskeleton reorganization, which involves the downregulation of epithelial markers (such as epithelial cell adhesion molecule (Ep-CAM), E-cadherin and keratin) and the overexpression of mesenchymal markers (such as vimentin and N-cadherin). This step occurs in a sequence: starting from a pure epithelial phenotype, evolving to a hybrid phenotype to, eventually, culminate in a pure mesenchymal phenotype, thus originating the heterogeneity of CTCs [18,22,23]. Indeed, CTCs are very heterogeneous, and different subpopulations may be found, including epithelial, mesenchymal, stem cell-like or mixed-cell profiles [30]. Additionally, these authors investigated metastatic renal cancer in 14 patients under anti-angiogenesis treatment and observed that the presence and quantity of N-cadherin-positive or CD133-positive CTC was associated with reduced progression-free survival. Another interesting finding was the inverse correlation between the presence of these CTCs and high expression of HIF1A, VEGFR, VEGFA and FGFR [30]. Using immunofluorescence in mouse models, Hassan et al. found hybrid CTCs, which were predominant in two models [31]. They also depicted a distinct gene expression profile in CTCs (high co-expression of epithelial and mesenchymal genes) compared to the primary tumor, thus identifying putative genetic signatures for hybrid CTCs or individual epithelial and mesenchymal CTCs [31]. EMT molecular markers also allowed for the classification of CTCs, identifying the most aggressive subpopulations [32]. For example, the presence of mesenchymal CTCs and circulating tumor microemboli (CTM) with a mesenchymal phenotype were more frequent in the metastatic stage than in the early disease stages in several types of cancers [33]. Moreover, mesenchymal CTCs have been linked to disease progression, whereas epithelial CTCs appear to be associated with response to treatment [19]. Nonetheless, contrasting results were found by Cappelletti and colleagues in an observational study. They collected twenty-one blood samples from ten metastatic RCC patients participating in the TARIBO trial and detected two different CTC subpopulations: one composed of epithelial CTCs (28% positive rate) and another without epithelial and leukocytic features, termed unconventional CTCs (62% positive rate). Intriguingly, they found a low progression-free survival (less than five months) in two patients with detectable epithelial CTCs at baseline, indicating that identification of a single of these CTCs before systemic treatment may be sufficient to predict short progression-free survival [34].

After morphological alteration is complete, cancer cells are now invasive, able to enter the blood vessels and disseminate along the circulation, enabling cancer progression [20]. Yet, once in the bloodstream, few CTCs (about 0.01%) survive due to the action of several factors such as anoikis, blood flow shear forces and the immune system that, ultimately, result in CTC death [21,22]. These few surviving CTCs may survive with the joint help of coagulation and inflammation, as recently found by Li Wen and colleagues. These authors studied CTC count as well as the levels of plasma C-reactive protein and fibrinogen of 106 RCC patients. They reported a positive association between these three variables, that is, an elevation of CTC count was found in patients with increased levels of both C-reactive protein and fibrinogen, suggesting a potential preventive role of these two systems in CTC death [35]. Very recently, platelets have been implicated in CTC resistance to cell death in the bloodstream [36]. T cell depletion has also been related to the immunosuppressive capacity in ccRCC tissues, promoting disease progression [37]. Interestingly, another cell type, circulating RCC-derived stem cells, appears to contribute to tumor evasion, as well [38]. After escaping death, CTCs must extravasate the bloodstream in the proper secondary location so they may colonize it and develop micro- and macrometastasis. To reach this goal, CTCs need to reverse EMT in a process called mesenchymal-to-epithelial transition (MET). During this process, CTCs decrease the expression of mesenchymal markers and increase the expression of epithelial markers, thus acquiring a proliferative ability to form metastasis [18,20,22]. This process was recently demonstrated in a study of a colorectal cancer patient [39]. It is also important to note that CTCs do not always directly metastasize following extravasation, as they may experience the not fully understood process of dormancy. This dormant state may last for many years (even decades) after a patient has had surgery to remove the primary tumor and being considered “disease-free”. In fact, during dormancy, CTCs are not destroyed by chemotherapy or the immune system and may remain quiescent until, under certain external stressors, they are activated and revert to a proliferative state, leading to late cancer recurrence [18,23,40,41,42,43].

Lately, different metastatic sites have been linked to different histological subtypes of metastatic RCC and to overall survival. For example, in a multicenter cohort study involving 10,105 renal cancer patients, ccRCC was predominantly associated with pancreatic, lung and adrenal metastases [44]. On the other hand, pRCC showed a higher propensity to metastasize to lymph nodes, and chrRCC showed more common liver involvement. In addition, metastatic localization was related to overall survival, with shorter survival depicted for bone, brain, pleural and liver involvement [44].

In particular, at early cancer stages patients may have CTCs in the bloodstream, in which they can travel singly or in clusters [21]. CTC clusters are characterized as groups of two or more CTCs that share stable cellular adhesions. These are named homotypic when composed only of cancer cells or heterotypic when they also integrate other components such as endothelial cells, platelets, leukocytes or fibroblasts [45]. It is noteworthy that CTC clusters are endowed with extreme survival in the bloodstream, have a much greater ability (about 23 to 50 times) to give rise to metastasis, and their presence is associated with worse clinical prognosis than individual CTCs [46]. Recently, the presence of a heterotypic cluster composed of CTCs and white blood cells was associated with lower overall survival compared to homotypic clusters in gastric cancer patients, suggesting a potential prognostic role of these clusters [47].

Generally, CTC clusters correspond to a minority of the CTCs present in circulation [46]. However, in mouse models, Suo and colleagues observed a relevant increase in the proportion of CTC clusters during the metastatic process, indicating a possible higher proportion of CTC clusters in the CTC population than previously expected [48].

Metastasis formation has been the subject of intense research over the years, and chromosomal changes have been gaining increasing importance in this process. Indeed, a prospective multicentric study of 100 patients with metastatic RCC evaluated the pathways of progression to metastasis [49]. The authors showed that the potential for metastasis is not due to the mutational burden of the driver genes, but rather to chromosomal complexity. In fact, the loss of 9p and 14q was shown to be the hallmark of genomic alterations for metastasis formation. Furthermore, the loss of 9p was also associated with the risk of mortality due to kidney cancer. Another curious fact was the existence of different patterns of metastatic spread. Specifically, there was a rapid progression to multiple sites in primary tumors of monoclonal structure, as opposed to an attenuated progression observed in cases with highly heterogeneous primary tumors. In the latter, metastatic potential was acquired gradually, initially evolving into solitary metastases. Intriguingly, pancreatic metastases evolved in a particular way, diverging early from the primitive clones and remaining in latency for around twenty years [49]. In another study, the same authors analyzed the evolution of ccRCC and found a highly conserved sequence of driving events that reveals its deterministic evolutionary nature [50]. Analyzing clonal dynamics, they found that early fixation of several driving events promotes accelerated tumor growth and metastatic progression. On the other hand, subclonal diversity was associated with slower growth and an attenuated pattern of metastasis. They concluded that chromosomal complexity and genetic variability had a significant impact on patient prognosis [50]. Along the same line of thought, Ye et al. analyzed 74 blood samples from RCC patients and identified triploid, tetraploid and polyploid CTCs for chromosome 8. They also observed that patients at stage III and IV disclosed higher CTC counts as well as more tetraploid, polyploid and large CTCs than patients at stage I and II, emphasizing the use of CTC enumeration and chromosome aneuploidy quantification as potential biomarkers impacting drug therapy efficacy and disease progression [51]. Regarding patients with inferior vena cava thrombus, triploidy of chromosomes 7 and 8 has recently been highlighted as the most frequently found karyotypes, with polyploidy being a rarity [52]. Beyond chromosomal changes, genetic changes have also been investigated at the molecular level. For example, Guan et al. performed a bioinformatics analysis of seven gene chip sets to identify different gene expression patterns between primary tumors and CTCs [53]. They detected 589 differentially expressed genes and found epithelial and mesenchymal CTCs in kidney cancer samples. At the same time, differentially expressed genes were particularly related to cell adhesion, epithelial–mesenchymal transition (EMT) and apoptosis in CTCs, inferring that CTCs mainly alter these variables during cancer progression. Intriguingly, *PSMC2* expression was increased in RCC tissues compared to normal tissues, and it was also associated with poor prognosis, leading the authors to hypothesize that *PSMC2* may impact RCC evolution and suggesting that PMSC2 is an indicator of poor clinical outcome [53]. Genetic differences between CTCs and primary tumors have also been identified in other cancers such as breast and bladder cancer, emphasizing the distinct genomic profiles of CTCs and their contribution to treatment response and follow-up [54,55].

A significant body of literature has proposed that stem cell-like CTCs, a subpopulation of CTCs with stem cell characteristics (such as self-renewal and differentiation into several cell types’ abilities), also play a role, promoting the progression of different cancers [42]. In that vein, Khan et al. conducted a study on RCC cell lines to assess the presence of stem cell-like cancer cells, specifically CD105+ cells (known as tumor-initiating cells), and compared their gene-expression profiles between primary tumor and metastasis. Notably, they found increased colonization ability as well as countless CD105+ subpopulations with higher expression of stemness-related genes (Oct-4 and Nanog) in metastatic compared to primary RCC cell lines. Moreover, CD105+ cells expressed mesenchymal stem cell markers (e.g., CD44, CD73, CD90, CD146 and alkaline phosphatase activity), whereas expression of CD24 or hematopoietic cell markers was absent. Compared with benign renal cell lines, CD105+ RCC lines differentially expressed 1411 genes, and the activation of certain transcriptional regulators (TGFB1, TNF and ERBB2) was most relevant in these cells. The authors then highlighted stem-like features of CD105+ cells and pointed out their exclusive gene-expression profile that might constitute novel therapeutic targets [56]. Subsequently, Shi et al. focused on the clinical value of CD105 expression among the three main RCC subtypes (ccRCC, pRCC and chrRCC) and its association with overall survival [57]. They recruited three cohorts of 604 cc RCC, 320 pRCC and 89 chr RCC and evaluated their respective outcome data. Then, bioinformatics analysis disclosed increased expression of CD105 in ccRCC compared to non-cancerous renal tissue. Interestingly, this overexpression only occurred in ccRCC, whereas the reverse was observed for chRCC and pRCC, leading the authors to hypothesize a relevant role of CD105 in ccRCC only. Contrasting results with previous studies were also found; namely, the high CD105 copy number in ccRCC correlated significantly with longer patient survival. Additionally, in ccRCC and pRCC, CD105 mRNA expression correlated with the presence of metastasis and tumor stage. They thus concluded that CD105 expression, through enhanced transcription of associated genes, potentiates RCC progression, although the association between CD105 and various stages of RCC is complex. Among the limitations of the study, the authors highlighted that CD105 also contributes to the activation of angiogenesis-associated factors, as well as the failure to consider the role of hypoxia in RCC and the lack of in vitro or animal model experiments to better sustain their hypotheses [57]. More recently, a study with RCC cell lines in mouse models investigated the ability of cancer stem cells to form tumors. The researchers found distinct subpopulations of these stem cells (based on expression of CD105, CD133, CD44 and CXCR4 markers), and the predominant one was the CD44+ subpopulation. Furthermore, tumor growth was observed in all subpopulations except CD105- and CD44+/CD105+ subpopulations. In general, the CD44-/CD105- subpopulation showed different metabolic profiles and a particular tumor evolution pattern, resulting in accelerated growth in only seven weeks compared to the other tumors, thus depicting greater aggressiveness [58].

## 4. Enrichment and Detection Techniques for CTCs

In this section, the techniques currently used for CTC enrichment and detection are described, and a brief overview on recently emerging techniques for RCC is provided.

As mentioned earlier, CTCs are rare cancer cells, difficult to find in the bloodstream. In fact, in the circulation, only about 1 CTC per 10^5^–10^7^ leukocytes is found in metastatic cancer patients, and this is expected to be lower—about 1 CTC per 10^8^ leukocytes—in non-disseminated cancer [22]. Moreover, CTCs are heterogeneous. These two factors combined make CTC enrichment and detection challenging, requiring highly sensitive, specific and reproducible technology [18].

In a simple way, technologies used for CTC isolation have two main objectives: enrichment and detection. Usually, CTCs are first enriched and then detected. Regarding enrichment, the main goal of this process is to concentrate CTCs as much as possible in a blood sample to allow for further analysis. Thus far, CTC enrichment may be accomplished through two categories of methodologies: label-dependent and label-independent. The first category receives this designation because it depends on cell surface markers [43]. It includes two types of selection: positive selection, which concerns the isolation of CTCs from the sample using tumor-associated epithelial markers, and negative selection, which reflects the extraction of leukocytes from the sample using specific markers present in these cells, thus selecting CTCs through the separation of contaminating blood cells. On the other hand, the second category is based not on cell surface markers but rather on the physical properties of CTCs such as size and deformability, density and differential electrical charge [43,59].

After the enrichment step, the CTC detection step follows. This includes techniques in which the primary targets are proteins (such as flow cytometry and immunofluorescence), and thus are mediated by antibodies, and others which focus on nucleic acids (such as reverse transcription polymerase chain reaction (RT-PCR) and real-time quantitative polymerase chain reaction (RT-qPCR), among others) [22].

## 5. Enrichment Techniques

### 5.1. Label-Dependent Methodologies

As mentioned above, these methodologies apply the principles of immuno-affinity, allowing CTCs to be isolated through the specific interaction that is established between antibodies and particular surface biomarkers [43].

#### 5.1.1. Positive Selection Strategy

This approach brings together several types of technologies that may fall into one of three distinct classes: immunomagnetic, microfluidic and combined devices. Immunomagnetic devices make use of monoclonal antibodies coupled to magnetic particles (such as iron) whose targets are surface antigens of CTCs (surface markers) and requires an incubation phase in a magnetic field. Microfluidic devices, on the other hand, permit the slow passage of blood on a chip-based surface that integrates antibodies coupled to microposts or to the surface, to which the specific antigens of the CTCs will bind, thus allowing for their isolation. Finally, the combined devices use both immunomagnetic and microfluidic principles in the same assay to facilitate the enrichment process [22,42].

As noted, positive selection starts through the binding of specific antibodies to CTCs’ surface markers, such as EpCAM, a transmembrane glycoprotein that has the advantage of being expressed by normal epithelial cells but absent in blood cells [22]. The same advantage is provided by another widely used epithelial marker, cytokeratins. Given the epithelial nature of RCC, these markers are particularly useful for respective CTCs enrichment [59].

One notable, and the most widely used, system in this domain is Cell Search^®^. This system is presently considered the gold standard for CTC enrichment, in addition to being the only system approved by the U.S. Food and Drug Administration (FDA) to enumerate CTCs in a metastatic setting [43]. It is based in an immunomagnetic technology, using antibodies against EpCAM conjugated to iron particles that interact with EpCAM surface antigen of CTCs, allowing for isolation when under a magnetic field [60]. Subsequently, the presence of EpCAM positive CTCs is verified by immunostaining using cytokeratins’ (CK 8, 18 and 19) expression, and the contaminating leukocytes are identified with the same method through CD45 expression [43,46]. This system also allows for image analysis using the CellTracks Analyser, which creates images of single CTCs that may then be reviewed and counted, resolving any mishaps that may impair CTC stability during the sample shipment and storage phases [42,61]. The CellSearch^®^ system has several advantages, allowing for semi-automated capture associated with CTC staining and counting, multiprocessing of samples (about eight, simultaneously), high level of reproducibility as well as efficiency [43,61]. However, its disadvantages are worth mentioning: the amount of blood collected limits the system, and it only detects CTCs with positive expression and increased levels of EpCAM, a feature which affects all methodologies which rely on EpCam staining [60,61]. This is of particular importance because CTCs have a dynamic nature, reducing their EpCAM expression when undergoing EMT. Therefore, the CellSearch system misses other CTC subtypes that do not express the EpCAM marker, promoting false negatives and underestimating the actual population of CTCs in the sample [46].

Recently, a study using the Cell Search^®^ system identified CTCs in about 60% of participating mRCC patients prior to tyrosine–kinase inhibitor (TKI) therapy, a higher percentage than that found in other mRCC studies with this system [62]. In addition, approximately half of the patients (46.7%) had at least one CTC in the blood samples whereas 15.4% of the patients had three or more CTCs, which are values inferior to those found in other solid neoplasms. Regarding EpCam marker expression, it was found that it may occur throughout disease progression, supporting the use of EpCam-based systems, of which Cell Search^®^ is an example [62].

Regarding microfluidic technologies, Liu et al. used the NanoVelcro platform to evaluate the detection efficiency of CTC, based on a new set of CTC surface markers, specific for RCC patients [63]. Two of these markers were carbonic anhydrase 9 (CAIX9) and CD147. They found a high efficiency of CTC capture in 10 RCC patients from the integrated approach of the NanoVelcro platform with anti-CAIX and anti-CD147 antibodies, compared to technology using the EpCAM marker only. In fact, CTCs were isolated in a percentage of 94.7% of RCC patients (72 out of 76 patients) with this technological approach. Furthermore, they suggested a possible prognostic value for CTC count and its associated vimentin expression (mesenchymal marker) in RCC patients due to a relevant association they found between these and RCC progression (including pathological stage) [63].

Another study involving 29 patients with metastatic ccRCC used, in addition to CAIX 9, the EpCAM marker in a new technology termed ESP [64]. This technology comprises a microfluidic-based platform coupled with anti-CAIX 9 and anti-EpCAM antibodies to provide high sensitivity in capturing CTCs. In addition to sensitivity, the authors achieved high specificity using a precise panel of exclusion cell markers (CD11b, CD14, CD34, CD45 or CD235ae antibodies) and detecting CTCs using fluorescence microscopy with CAXII and CK protein staining. Using ESP technology, the authors found significant heterogeneity of CTCs in the blood of patients with metastatic ccRCC, which was reflected in the detection of subpopulations of CTCs expressing distinct combinations of the markers CAIX 9, EpCAM, CAXII and CK. Among the advantages of this new technology is the possibility to sequentially perform several purification processes, as well as the high recovery rate and reduced cell loss [64].

#### 5.1.2. Negative Selection Strategy

As opposed to positive selection, this approach is based on removing contaminating leukocytes, taking into account the expression of surface markers on these white blood cells [22]. CD45 is the marker of choice, being an antigen present on all leukocytes [59]. Like positive selection, negative selection uses immunomagnetic devices and takes advantage of the antibody–antigen interaction (in this case, CD45) to trap leukocytes as they pass through magnetic beads impregnated with anti-CD45 antibodies [61]. Since no CTC surface markers are used, negative selection allows for heterogeneous CTC populations (epithelial, hybrid, mesenchymal and others) to be obtained, including CTCs that show reduced EpCAM expression, and this is an important advantage [22,61]. However, there are also disadvantages to be taken into account, namely, the lower purity rate compared to positive selection [60] also promoted by the loss of CTC-WBC clusters that are removed along with the leukocytes and the occasional loss of CTCs that are inadvertently retained with the blood cells and are removed along with them [61]. Nonetheless, several studies have used this type of immunomagnetic enrichment in the field of RCC-CTCs in recent years [52,65]. For example, Zhang et al. conducted a study in RCC patients with inferior vena cava thrombus using this approach combined with immunocytochemical staining for the first time [52]. They initially performed a hemolysis process on 20 blood samples followed by centrifugation, using the cell pellet obtained for the negative selection itself. For this, a 30-min incubation period of the sediment in magnetic beads conjugated with anti-CD45 monoclonal antibodies was necessary, being later separated with the help of a magnetic separation support (from Promega). Using this negative selection technique, it was possible to remove contaminating CD45+ leukocytes and facilitate CTC concentration in the samples for further identification of the supernatant layer [52].

### 5.2. Label-Independent Methodologies

This category of methodologies is an alternative option to CTC enrichment based on surface biomarker expression, which, as acknowledged, is highly variable among CTCs, with some of them subjected to marked downregulation, not even expressing EpCAM and/or CK markers. Thus, it is a good strategy to isolate CTCs that undergo EMT. It includes technologies based on the physical characteristics of CTCs, namely size and deformability, density and differential electric charge [66].

#### 5.2.1. Size- and Deformability-Based Enrichment

This is a method that takes advantage of the principle that CTCs have larger dimensions than blood cells and are less able to deform. It includes several types of technologies, including membrane microfilters and microfluidic chips, among others [43].

##### Membrane Filtration

Membrane microfiltration technology includes equipment consisting of a membrane that functions as a filter having pores of specific shape and size (between 6 and 9 µm). Using a centrifuge or pressure regulator, the blood passes through this membrane and the smaller blood cells are filtered out, isolating CTCs, without any contribution of cell surface markers [61,66]. Among the many advantages of this technology are the ability to collect more representative phenotypic populations of CTCs and reduce cell loss, as well as the general speed and ease of production associated with low cost [61]. However, there are some limitations to mention, namely the need for higher volumes of blood (approximately 10 mL) for its application, the high probability of resistance to filtration due to pore clogging with cell agglomeration and the loss of smaller CTCs compared to pore size [59,66,67]. One example of these technologies is the Isolation by Tumor Cell Size, better known as ISET^®^ system. This system differs from others of its type due to its longer processing time (about 4 h to process 10 mL of blood) and the need for a 1:10 blood dilution to prevent membrane clogging. It is composed of 8 µm-diameter cylindrical pores and allows the vacuum-assisted filtration of several samples (about 12) simultaneously [61]. Interestingly, Bai et al. wanted to compare the detection efficiency of this system with the FDA-approved CellSearch^®^ system in RCC patients. To accomplish this, they enrolled 36 RCC patients and 22 healthy individuals and analyzed the samples on both systems [68]. The specificity achieved was 100% for both systems, with the total absence of CTCs detected in healthy individuals. On the contrary, the overall ability to detect CTCs in RCC patients was higher for the ISET system 36.1%, with CTCs detected in 13 of 36 patients and circulating tumor microemboli (CTM) detection in 3 patients, compared to the CellSearch system (19.4%, with CTCs detected in 7 of 36 patients). Considering also that ISET system was able to detect CTMs, the authors concluded the superiority of the ISET system over the CellSearch system in detecting RCC-CTCs [68].

Recently, a study involving multiple cancer types, including RCC, compared two capture platforms: one based on the slit filter (CTC-FIND) and another based on free selection (AccuCyte^®^-CyteFinder^®^ system) [69]. A significantly higher positive CTC rate (91.7%) was found using the CTC-FIND platform compared to the AccuCyte^®^–CyteFinder^®^ system (66.7%). Furthermore, the median diameters obtained from CTCs of RCC patients using the CTC-FIND platform were significantly higher. It was concluded that the CTC-FIND platform has a superior ability to capture positive CTCs, however of larger size. A warning was issued concerning the bias related to a higher preponderance of prostate cancer cases contributing to the results obtained and the small size of the sample used [69].

CanPatrol CTC™ system is another membrane filtration-based device used in several studies [70,71,72]. In addition to CTCs, this system is also able to isolate CTC-WBC clusters [70].

Besides CTC enrichment, size-based methods may benefit CTC cultures. One such method integrates the MetaCell^®^ technology by filtration, which constitutes a useful tool to isolate CTCs larger than 8 μm for subsequent in vitro cultures, thus facilitating efficient CTC transfer to the culture site. Using this technology in a protocol, it was shown that among CTCs from genitourinary cancers, the largest size is found in RCC-CTCs [73].

##### Microfluidic Chips

The size-based microfluidic chip is also called “three-dimensional microfiltration”, as it uses 3D geometries that create specific spaces for CTCs and non-CTC cells, promoting the isolation of CTCs from the remaining blood cells. An illustrative example of this technology is the Parsotix^®^ system. This system has a trapezoidal conformation, and its width is gradually reduced (to about 4.5 µm) to isolate the trapped CTCs and remove the smaller cells that cross the channel. In addition, the isolated cells are protected from deformation thanks to a protective physical support that surrounds them [43,66]. Like filtration membranes, microfluidic chips allow for the isolation of heterogeneous CTC populations; however, CTC recovery rates are low, in addition to the fact that some contamination of similarly sized WBCs may occur [61].

In 2022, a study conducted by De Alwis et al. involving 16 patients with RCC and several RCC cell lines used the microfluidic-based enrichment technology ClearCell FX system [74]. This system departs from the premise that CTCs have larger dimensions than contaminating leukocytes, which was proven in the study by evaluating cell diameters. Moreover, CTCs presented diameters greater than 18 µm, above the lower size limit (14 µm) of the system used, making it appropriate for the isolation of CTCs from RCC. With the aid of reverse immunofluorescence microscopy, the authors evaluated the number of CTCs and obtained a CTC recovery efficiency of 66% using the Cell ClearCell FX system, higher than those of other enrichment systems (such as CellSearch^®^) [74].

#### 5.2.2. Density Based Enrichment

Density-based technologies use centrifugation combined with different density gradients to separate the various blood ingredients (mononuclear cells, polymorphonuclear cells and red blood cells). Initially, blood is diluted and then deposited into a density gradient medium and centrifuged. In this way, cells are distributed according to their sedimentation coefficients, with cells with higher density sedimenting at the bottom of the tube (erythrocytes and leukocytes) while those with lower density (such as platelets and mononuclear cells, such as CTCs) remain at the top, in an interphase layer, allowing their isolation for later analysis [22,43]. As may be deduced, one of the inherent problems with this isolation technique is the low efficiency and purity rate due to losses of higher-density CTC and CTC aggregates that sediment in the bottom layer, leading to false negatives. However, it is a low-cost and simple technique to apply and was one of the first techniques initially employed in CTCs’ isolation [59,61,66].

#### 5.2.3. Differential Electric Charge-Based Enrichment

Dielectrophoresis technologies take advantage of differential electrical charges to separate CTCs from contaminating leukocytes. These different electrical characteristics between cells depend on cell polarity and are influenced by several factors, including cell density, size, volume and conductivity. Indeed, these technologies allow CTCs to be moved distinctively with the help of dielectrophoretic field strengths, selecting single CTCs with high accuracy, specificity and viability, independent of the expression of biomarkers and other blood cell components [43,59,61,66]. In addition, dielectrophoresis technologies allow for the identification of CTCs with various phenotypes for subsequent functional analysis and reduce operator dependence, as well as cost and time spent, compared to label-dependent technologies [75]. Like the other enrichment techniques, it also has disadvantages, namely its application in a restricted volume combined with the occurrence of possible electrical changes in the cells throughout the process as well as the need to monitor several factors at the same time [66].

### 5.3. Detection Techniques

#### 5.3.1. Protein Expression-Based Strategies

##### Flow Cytometry

Flow cytometry allows for quantitative protein-based assessment of CTCs and is useful in both their detection and characterization. This technique uses specific antibodies with fluorescent properties to label cells that will pass through a flow cytometry column. Thus, as they pass through this column, characteristics such as fluorescence excitation and light scattering inherent to each cell provide relevant information (namely size, shape and presence of granules, among others) to classify CTCs. Moreover, flow cytometry enables evaluation of the expression of multiple markers and, thus, characterizes CTCs more adequately. Other positive points to note about this technique are the easy determination of the existence or absence of proteins, the quantification of protein expression grades among CTC subpopulations as well as the possibility to collect these subpopulations. Downsides include the impossibility of visual confirmation of labelled CTCs as de facto CTCs and the low sensitivity for scarce CTC populations [22].

Recently, a study applied this technique to detect CTCs using cell staining based on a set of extracellular protein markers that included two ccRCC-specific markers (CAIX and CAXII), EpCAM and CK. High heterogeneity of ccRCC-CTC marker expression was found, and CTC subpopulations with different degrees of co-expression of these markers were reported in different patients. Notably, expression of the markers CAIX and CAXII was observed in all samples analyzed, disclosing higher frequency than EpCAM marker expression [64]. Another study enrolling 533 ccRCC patients used a “digital” cytometry technology—CIBERSORTx—to detect CTC clusters. Indeed, the authors identified multiple cell types (such as CTCs, fibroblasts, CD8+ and endothelial cells) via single-cell RNA sequencing information, illustrating intratumoral heterogeneity in ccRCC [37].

It is noteworthy that flow cytometry strategies have been used in other neoplasms, such as breast cancer and head and neck squamous cell carcinoma. In 2022, Staudte et al. used a technology termed Amnis^®^ brand ImageStream^®^X MkII (ISX) to perform a multiparametric phenotype analysis on CTCs. Promising results were depicted regarding programmed death-ligand 1 (PD–L1) expression, EGFR activation and DNA damage repair in CTCs. Furthermore, this technology has been revealed to be sensitive (73%) and highly specific (100%), using a cut-off value of ≥3 CTCs [76].

#### 5.3.2. Immunocytochemistry

##### Immunofluorescence

Immunocytochemistry includes the immunofluorescence technique, which uses the labelling of target antibodies with fluorophores, allowing for the observation of CTCs’ proteins by fluorescence microscopy. Indeed, CTCs may be characterized by certain microscopic and differential staining patterns. In terms of a protein-based strategy, immunofluorescence is the most frequently used technique, also enabling further characterization of CTCs [22]. An example of the use of this technique was performed by Liu et al. [63]. Various cell markers, such as DAPI, anti-CD45, fluorescence-labeled anti-pan-CK and anti-vimentin antibodies were used to identify different cell types. Through the distinct expression of these markers, the authors were able to detect three cell types, two of which were CTCs, characterized by DAPI+/CK+/CD45-/Vimentin+ expression and DAPI+/CK+/CD45-/Vimentin- expression [63]. In addition to CTC characterization, immunofluorescence has other advantages, such as the possibility to visually verify both protein expression and localization through the associated fluorescence as well as the qualitative assessment of different levels of protein expression. Furthermore, there is the possibility to analyze several proteins at the same time with several fluorescent filters. It discloses several drawbacks, such as other techniques, including the difficulty in having specific antibodies available or creating new ones, low sensitivity and imprecision of interpretation due to autofluorescence of the cells [22].

#### 5.3.3. Nucleic Acids-Based Strategies

##### PCR

Very recently, a study used digital PCR to detect RCC-derived CTCs. This was performed at the molecular level, after extraction of RNA from previously enriched CTCs and subsequent global reverse transcription pre-amplification reaction of the cDNA to expand 400–10,000x the number of fragments, allowing for multiple analysis. The authors evaluated information from the TCGA (The Cancer Genome Atlas) database and identified ccRCC-specific genes. Using the distinct expression of those genes (CA9 and SLC6A3) in digital PCR, the presence of ccRCCs, even at the single cell level, was disclosed. Interestingly, the detection was more reliable with the CA9 gene marker than SLC6A3, with a greater consistency of results [74].

##### RNA In Situ Hybridization

This technique allows for the various types of CTCs to be identified based on RNA expression. After an amplification phase, several RNA probes are hybridized with fluorescent probes, enabling the identification of specific genes of mesenchymal (TWIST and VIM) or epithelial (EpCAM, CK8, CK18 and CK19) lineage through the different fluorescence emitted. Using this technique, it is possible to typify mesenchymal, epithelial and hybrid CTCs as well as CTC clusters [70]. Recently, a study including 69 RCC patients used this technique to determine the expression of Beclin-1, a gene related to autophagy, in RCC-CTCs. Interestingly, a significantly higher count of CTCs expressing Beclin-1 was reported compared to CTCs not expressing Beclin-1, before surgery, in the mRCC patients, leading to the inference that the expression of this gene may be implicated with metastatic activity. In any case, shortcomings of the study such as the small number of participants and a short follow-up period were highlighted [72].

##### Sequencing

Sequencing integrates several methodologies that have, as a common point, to obtain information at the genome level or at a certain genetic target region through the sequencing of nucleotide bases. It is a useful approach in both CTC detection and characterization, allowing unique genomic alterations present in DNA from the genome or from transcribed RNA sequences to be assessed. Positive aspects of this type of technique include automation, easy analysis of the results compared to immunofluorescence and the possibility to detect nucleotide aberrations at a single base level with potential cellular phenotypic impact. Downsides include the minimum limit of the number of CTCs for single-cell analysis and the possible existence of false positives or negatives due to the lack of visual verification of the origin of the amplified transcripts [22]. Recently, several studies used single-cell genetic analysis of CTCs from ccRCC patients, which confirmed the identification of CTCs based on cytomorphology and provided information about the intratumoral environment in ccRCC [37,77]. Next-generation sequencing (NGS) analysis has been also used in RCC genetic research. In 2021, a study used NGS to detect gene mutations with prognostic potential present in the circulating DNA of RCC patients [78].

#### 5.3.4. Combined Strategies

To overcome several disadvantages associated with different isolation technologies and to achieve better CTC recovery rates, researchers have chosen to combine various strategies. For example, a recent study isolated CTCs following a sequence of steps that included density-based devices (gradient centrifugation), negative selection and techniques for assessing the expression of EpCam and cytokeratin markers. Using this multi-step approach, a higher percentage of detection was obtained than that found with the Cell Search system. In addition, the presence of CTCs was found in 22.9% of the mRCC cases [79].

### 5.4. Emerging Techniques for mRCC

New technologies have been created in the field of mRCC, promoting better CTC detection and characterization (summarized in Table 1). One such technology is SE-iFISH, which consists of a combination of subtraction enrichment (SE) and immunofluorescence in situ hybridization (iFISH), being used in many studies [65,80,81]. One study found that CTCs were present in 70% of patients, and a 76.92% detection rate of CTCs was depicted in several genitourinary cancers [81]. Tian et al. applied this technology to detect CTCs in RCC patients and obtained a CTC positivity percentage of 86.20%. Besides the superior detection performance of SE-iFISH compared to other devices, this technology characterizes CTCs at both the phenotypic and karyotypic level (aneuploidy of chromosome 8) and allows for the detection of PD-L1 expression in RCC-CTCs [65,80]. It has also been shown to be highly specific and sensitive, making it suitable for RCC-CTC detection [65].

Similarly, Zhang et al. used iFISH for the identification of RCC-CTCs after a negative enrichment (NE) step in RCC patients with inferior vena cava (IVC) thrombus [52]. In this termed NE-iFISH strategy, immunomagnetic beads coated with anti-CD45 monoclonal antibodies were used for NE. Subsequently, specific iFISH probes were used to detect aneuploidy of chromosomes 7 and 8, present in RCC-CTCs. This size- and EpCAM marker-independent approach allowed for differentiating CTCs and assessing chromosomal aneuploidies. Therefore, simultaneous quantitative and qualitative evaluation may be performed based on NE-iFISH, with positive perspectives in RCC-IVC patients with vena cava thrombus [52].

Another new technology for RCC-CTC study is the tapered-slit filter platform, which relies on a strategy based on size and deformability. It allows for CTC detection efficiency of approximately 62%. These cells are most often detected in advanced stages of disease, according to Kim et al. [82]. Additionally, this platform may assist in prognosis assessment of RCC patients, since the reported survival (both cancer-specific and PFS) seems to be reduced in the presence of CTCs [82]. Regarding emerging size-based technologies, CTC size seems to be a relevant aspect to consider in their development, considering the differences in dimension of tumor cells from cell lines and circulating ones, as demonstrated in a large cohort study involving several cancer types [17].

Naoe et al. reported high accuracy (about 95%) in detecting RCC-CTCs using a novel approach integrating several technologies: CelSee^®^ and On-chip Sort^®^ combined with anti-G250 antibodies [83]. In a first step, CTCs were enriched using the CelSee^®^ microfluidic-based platform (label-free) and then analyzed and sorted using On-chip Sort^®^. An innovative aspect was the use of triple immunostaining with anti-EpCAM, anti-CD45 and anti-G250 (kidney cancer specific), followed by evaluation with flow cytometry using the BD FACSCalibur™ system. Compared to anti-EpCAM antibody, anti-G250 antibody shows a higher sensitivity for RCC-CTC, and there is a higher staining affinity for these specific cell lines than in other urological cancers (such as those of bladder and prostate). CTC analysis may be indicative of hematogenous spread, making this CTC detection approach suitable for evaluating cases with distant metastasis. Moreover, this approach may go further in terms of monitoring treatment and measuring drug efficacy, thus having an impact on mRCC therapy selection, as demonstrated in a case in which CTC count entailed the change of the initially prescribed drug [83].

In recent years, the use of nanomaterial-based technologies for CTC analysis in RCC studies has been growing. Silver nanoparticles coated with a silica shell were combined with a photovoltaic technology (SERS platform) integrated into a microfluidic device to analyze renal cell lines. The use of Raman spectroscopy (SHINERS) allowed this combination to result in label-independent separation and detection of RCC-CTCs with high sensitivity and accuracy (approximately 89%) [84]. Another study used nanobiointerface chips composed of silica for CTC detection in urological cancers. Remarkably, these chips showed high efficiency (approximately 80%) in identifying CTCs from prostate cancer and ccRCC, with low cost [85]. Besides nanomaterials, surface engineering has also been striving to invest in new platforms for capturing RCC-CTCs. In 2020, a new platform was created, incorporating various strategies such as dendrimers in the surface coating, biomimetic cell rolling and antibody cocktail (anti-EpCAM, anti-CA9, anti-EGFR and anti-c-MET). Using this platform and the inherent antibody cocktail, an effective CTC capture was achieved, significantly exceeding the CTC counts obtained with the analysis based on EpCAM expression alone [86].

Cytopathological evaluation of RCC-CTCs also raises researchers’ interest. This is highlighted in a recent study that combined label-free technologies for the enrichment of CTCs with pathological study using hydrogel-based cell blocks [87]. Briefly, the cell block construction included several cancer cell lines, and immunocytochemistry was used in screening for potential CTCs markers. After isolation based on physical properties, CTCs were assessed by immunocytochemistry for the specific markers that had been tested in the cell blocks and would move on to the pathological study. This versatile approach may revolutionize the way specific CTCs are detected (label-free) and might be applicable to any cancer in the emergence of new biomarkers [87].

Currently, in vivo methods have emerged as forms of CTC detection that overcome the constraints of ex vivo analysis, namely blood volume. These include the CellCollector^®^ technology, that not only effectively isolates RCC-CTCs in vivo but also ex vivo, as recently demonstrated by Bialek et al. [12]. This is a technology based on EpCAM expression, using a medical thread coated with anti-EpCAM antibodies. Bialek et al. used straight and spiral wires coated with anti-EpCAM and anti-MUC1 antibodies in their analysis. Interestingly, the addition of the anti-MUC1 antibody enabled not only RCC-CTCs to be identified but also patients with lung metastasis. Therefore, MUC1 (a surface protein) was suggested as a molecular marker predictive of lung dissemination [12].

The main steps involved in RCC-CTC processing are graphically illustrated in Figure 2.

**Table 1 cancers-15-00287-t001:** Summary of the main emerging technologies applied in mRCC-CTCs.

Technology	Characteristics	Reference
SE- iFISH	Based on subtraction enrichment (SE) and combined with immunofluorescence in situ hybridization (iFISH)	[65,80,81]
NE-iFISH	Based on negative enrichment (NE) and combined with immunofluorescence in situ hybridization (iFISH)	[51]
Tapered-slit filter platform	Based on size and deformability properties	[82]
CelSee^®^ + On-chip Sort^®^ + BD FACSCalibur™ system	Based on microfluidics and sorted by combination with anti-G250 antibodies Based on flow cytometry detection	[83]
SERS platform and Raman spectroscopy (SHINERS)	Based on nanomaterials (silver) combined with photovoltaic and microfluidic devices and spectroscopy	[84]
Nanobiointerface chips	Based on nanomaterials (silica) combined with nanochips	[85]
Capture platform	Based on dendrimers, biomimetic cell rolling and a combination of antibodies	[86]
Hydrogel-based cell blocks	Based on cell blocks formation (label-free) combined with immunocytochemistry	[87]
CellCollector^®^	Based on EpCam expression In vivo and ex vivo method	[12]

## 6. Models for RCC Investigation

Besides liquid biopsy, RCC- CTCs may be also analyzed using several different models which can be allocated to two main groups: in vivo and ex vivo models. The former includes microfluidic technologies, two-dimensional cell line cultures and three-dimensional organoids. Concerning the latter, they make use of animal models for xenografts and genetic engineering. Overall, animal models play a central role in the study of tumor and metastatic progression, provide more robust information on tumor microenvironment and allow for personalized drug evaluation, notwithstanding the associated high costs [88].

## 7. Clinical Applications of CTCs

The main clinical applications of CTCs are graphically summatized in Figure 3.

### 7.1. Early Detection of Metastasis

The scientific community has been focused on the ability of CTCs in early prediction of metastasis. In this regard, a study evaluated the relationship between the CTC-positive status of patients with mRCC and the expression of blood tumor markers, namely cytokeratin 19 (CK19), endoglin (CD105) and CD146 [89]. The measurements were taken at different time points, namely before and after surgical treatment. Interestingly, a statistically significant difference in the expression of CK19 and CD105 markers was observed in CTC-positive, compared to CTC-negative patients between measurements taken one day before and one week after surgery. The same was not observed for CD146. Another statistically significant difference was depicted for expression of the three markers between measurements taken one day before and one month after surgery. Thus, it was hypothesized that those markers, when in the presence of CTCs, may help in early metastases detection and prognosis assessment [89].

**Figure 3 cancers-15-00287-f003:**
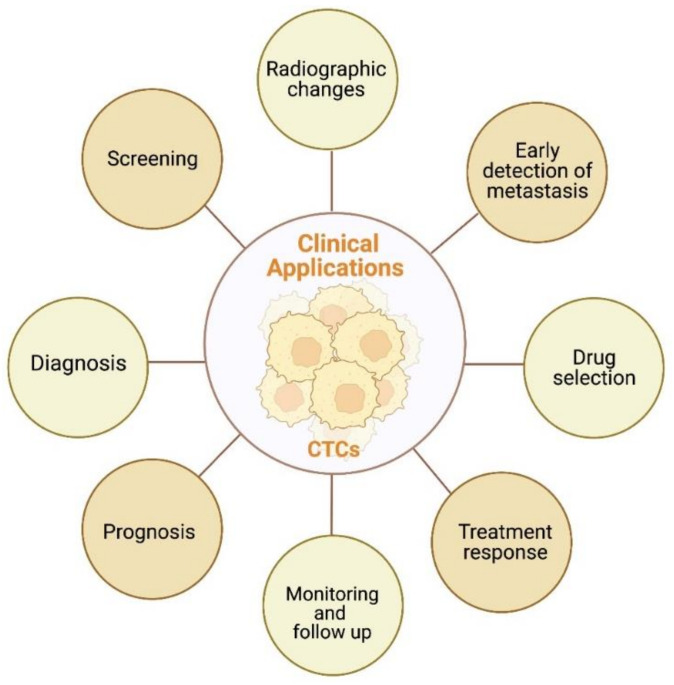
Schematic representation of potential clinical applications of CTCs in RCC. Created with BioRender.com.

During the metastatic cascade, CTCs interact with various blood cells, including platelets and inflammatory cells. In 2022, a study evaluated the prognostic impact of this set of cells in 82 RCC patients [36]. Together, the presence of mesenchymal CTCs, monocyte/neutrophil ratio (MNR) and staging demonstrated prognostic potential, allowing metastization after surgical treatment to be predicted [36]. In the same year, another study, enrolling 1176 nephrectomized patients, emphasized the relevance of long-term follow-up to detect early recurrence, as it seems to be a frequent event. In fact, about one-fifth of patients with stage I RCC experienced late recurrence (after 20 years of follow-up), and this figure was higher in advanced stages [90]. Intriguingly, in a retrospective study of 464 nephrectomized patients for non-metastatic RCC, shorter survival (for both cancer-specific (CSS) and overall survival (OS)) was observed in patients with local recurrence and absence of distant metastasis, thus reflecting a worse prognosis [91].

### 7.2. Prognostic Value

As for the prognostic value of CTCs, it is known that their presence in the bloodstream is associated with poorer prognosis [92]. In this context, Klezl et al. investigated 186 patients with RCC, assessing the presence of CTCs before and after surgery as well as during follow-up, using mitochondrial activity to infer about their viability. They found CTCs in 86.7% of the samples and noted that detection of CTCs was associated with tumor size and that it was more likely to occur when the tumor was growing. Tumor size was also associated with the presence of metastases in lymph nodes and distant sites [93]. In another study, the numbers of RCC-CTCs before surgery were found to be significantly associated with tumor size but not other clinicopathological features [94]. Moreover, there was a significant positive association between high Ki-67 expression (a proliferation indicator) and the numbers of CTCs before surgery, especially for a number greater than or equal to five CTCs. Thus, the presence of ≥5 CTCs was proposed as a diagnostic and prognostic parameter for RCC. Then, it was concluded that the surgical strategy, namely nephrectomy, is responsible for changing the levels of CTCs, directly leading to a postoperative reduction in their number in RCC patients [94].

In the same perspective, to evaluate the impact of surgical technique on CTC numbers, Haga et al. conducted a study that included 60 RCC patients [95]. Patients who underwent open radical nephrectomy disclosed significantly higher CTC numbers after surgery, as well as significant perioperative change in CTCs compared to patients who underwent laparoscopic or partial open surgery. In particular, the CTC numbers after surgery were significantly associated with larger tumor masses. Regarding preoperative CTC number, a significant association was found between tumor diameter and TNM staging, with the highest numbers seen in advanced disease stages. Finally, the authors pointed out the importance of surgical technique and tumor diameter in the spread of CTCs, which may promote early postoperative metastasis, thus affecting prognosis [95]. Another study pointed out a significant association between the presence in the bloodstream of six or more CTCs after surgery and RCC progression [96]. Moreover, the presence in circulation of mesenchymal CTCs and CTC-WBCs post-surgery correlated with higher likelihood of recurrence and metastasis. Hence, these three CTC subtypes were highlighted as independent prognostic factors of worse clinical outcome and shorter PFS [96].

Recently, an observational study looked at the ability of CTCs to predict invasion of the inferior vena cava by RCC. It was then hypothesized that the total CTC count as well as the CTC subtype could be predictive indicators of that invasion, having potential to influence RCC prognosis [80].

In a prospective multicenter study including 195 patients with mRCC, CTC counts were obtained and compared according to patient outcome [62]. On the one hand, an association was initially depicted (independently of the International Metastatic RCC Database Consortium (IMDC) score) between the presence of three or more CTCs, and on the other, reduced PFS (median 8.3 months vs. 32.6 months) and OS (13.8 months vs. 52.8 months). Hence, a cut-off value of three CTCs, using the Cell Search system, was suggested as having prognostic value. Furthermore, a correlation between higher propensity to CTC counts less than three, and bone involvement was depicted. Interestingly, the same was not found for brain and liver involvement [62]. In another study, a significant association between the presence of CTCs and shorter PFS was reported. Nonetheless, the same was not verified for OS. Thus, it was suggested that the positivity of CTCs at baseline may constitute a relevant prognostic factor in mRCC patients for PFS only [79]. Intriguingly, a study indicated an absence of correlation between the initial numbers of CTCs and the likelihood of RCC recurrence or metastization. Instead, it seems that it is the dynamic change in CTCs (particularly mesenchymal CTCs and those expressing Beclin1) that correlates with recurrence or metastasis. Still, several limitations of the study hamper the conclusions [72].

Different CTC subtypes appear to influence cancer progression in RCC patients. This is illustrated by a recent study in ccRCC patients, in which a significant association between the enumeration of CTCs and positive CK marker expression, and disease progression was depicted. Surprisingly, the same was not disclosed for CTC counts with unique expression of CAXII, a kidney cancer-specific capture antibody [64].

In addition to CTCs, CTC-WBC clusters have also been the subject of intense research. In a study involving 163 patients with RCC, the prognostic role of both CTCs and CTC-WBC clusters was evaluated [70]. High total CTC counts and the presence of CTC-WBC clusters disclosed prognostic impact for MFS, the latter being more crucial for a worse outcome. Importantly, tumor mass diameter together with high total CTC count demonstrated value as prognostic indicators for OS, enabling better outcome prediction in RCC patients [70].

Karyotype analysis in renal CTCs has sparked increasing interest among investigators. An interesting finding in terms of clinical application was the recent indication of tetraploidy of chromosome 8 as potentially prognostic in RCC. This was suggested by the significantly higher presence of tetraploid CTCs in the advanced stage of the disease (T4 stage) compared to the other stages [65]. Interestingly, epigenetic factors, such as DNA methylation, have also been investigated as ccRCC prognostic biomarkers. Indeed, their combination with clinicopathological features seems to provide a more accurate prognostication in patients with localized ccRCC [97]. Methylated DNA has also been recently characterized in CTCs from other urological cancers, such as prostate cancer [98].

Importantly, patient characteristics seem to impact CTC enumeration. A recent study demonstrated that age of RCC patients may interfere with CTC numbers, being negatively associated with survival. Moreover, the number of CTCs may be affected by the expression of Beclin1 in mesenchymal and epithelial RCC-CTC, which decreases its production. Intriguingly, in a study that evaluated Beclin1 expression in CTCs and CTC count, no differences in OS and disease-free survival (DFS) were apparent [71].

Besides the potential prognostic value of CTCs, other biomarkers have also been tested, and circulating tumor DNA (ctDNA) has been shown to be a worthy competitor [99]. In fact, higher ctDNA levels have been found in disseminated disease (as opposed to localized disease), and its detection in blood samples may be associated with worse clinical outcome (i.e., higher risk of death and lower overall survival). Moreover, variations in ctDNA levels along disease progression may provide valuable information for monitoring treatment response and follow-up of RCC patients. Interestingly, the detection of genetic aberrations in ctDNA during cancer treatment may be useful for deciphering mechanisms of resistance to therapy. However, downsides of ctDNA include the low levels in the bloodstream of RCC patients. Furthermore, the paucity of studies (and with small sample size) in RCC coupled with the low sensitivity of ctDNA detection methods used mean that there is still a long way to go regarding this biomarker, making CTCs more attractive [99].

### 7.3. Prediction of Response to Therapy

A study conducted by Basso et al. evaluated the response to TKI treatment in mRCC patients by CTC enumeration [62]. After approximately 10 months of treatment, with measurement of CTCs at various time points, there was no significant association between CTC numbers (either at baseline or in sequence, using cutoff values of one/two/three/four/five CTCs) and response to treatment. Moreover, regarding radiological evolution of mRCC patients, it was not possible to predict radiological outcome based on CTC numbers (either at baseline or dynamic) [62]. Nonetheless, contrasting results were disclosed in another investigation that indicated a significant predictive role for the presence of CTCs at baseline in weaker response to TKI treatment in mRCC patients. Thus, CTCs may be useful as predictive biomarkers of response to TKI therapy in mRCC [79]. More recently, another study indicated CTC counts as a pharmacodynamic biomarker, going beyond treatment with TKIs [64].

Promising results have also been recently reported concerning the therapeutic response, namely, a significant association with PD-L1 and HLA-I expression in CTCs. CTCs with positive expression for CAXII and PD-L1 were associated with response to immune-checkpoint inhibitors (ICI) treatment, whereas CTCs with positive expression for CAX II and HLA-I were significantly associated with response to TKI therapy. The same was not observed for CTCs with positive CK expression, and it was suggested that the various subpopulations of CTCs influence RCC patient response to therapy [64].

In another study including 155 patients with various types of cancer, PD-L1 expression in CTCs was proposed as a possible biomarker for response to immunotherapy [100]. In fact, elevated PD-L1 expression in CTCs was found to be both predictive (with higher response rate) and prognostic (higher progression-free survival and overall survival) in advanced stage cancers under immune checkpoint inhibitor therapy [100]. In particular, during treatment of ccRCC patients with PD-L1 inhibitors, dynamic changes in CTCs’ PD-L1 expression may have prognostic value, and CTC counts may play a role in treatment monitoring [101]. Concordant results were found in urothelial carcinoma, and the presence of CTCs expressing PD-L1 at baseline may further allow real-time selection of the most suitable candidates for immunotherapy [102]. Besides the role of PD-L1, other immunosuppressive checkpoints, such as TIM-3 and LAG3, were also pointed out as useful markers in the context of immunotherapy for ccRCC. Indeed, blocking these checkpoints may favor the prognosis of ccRCC patients, since depletion of immune cells, namely CD8+ cells, was associated with worse clinical outcome in a recent report based on transcriptomic analysis [37].

Besides the role of predicting response to therapy, CTC counts may be used to monitor therapy over time in patients with mRCC [103]. This is well illustrated by a 2022 longitudinal study involving 104 patients with mRCC, which reported a lower overall survival for patients who submitted to immunotherapy, with a trend for a higher number of CTCs over time [104]. In addition, a worse outcome was found in patients with a trend for higher values of the HLA I to PD-L1 ratio. Thus, it was pointed out that the change in CTC count and in the molecular profile of CTCs along treatment may serve as biomarkers for monitoring treatment response [104].

### 7.4. Screening

To date, scarce research has been performed on the potential role of RCC-CTCs in screening. In fact, one of the few existing studies has focused on combining CTCs with polymorphisms of cancer susceptibility genes, namely XPC A2815C and XRCC1 G1196A [81]. Surprisingly, this combination proved to be highly sensitive (92.86% positivity) in detecting genitourinary cancer patients. Furthermore, a direct relationship was established between CTC count and cancer progression, in which RCC was included. Therefore, the application of this panel in the screening and monitoring of genitourinary cancers, such as RCC, has been suggested [81].

### 7.5. Monitoring and Follow-Up

Ultimately, it has been proposed that the presence of high numbers of RCC-CTCs before surgical treatment (≥5 CTCs) should motivate a stricter monitoring and longer-term follow-up among these RCC patients [95].

### 7.6. Radiological Impact

In a recent study, an association between changes in numbers of different CTC subpopulations and radiological tumoral findings was reported. Particularly, variations in CTC CK+ numbers were related to radiographic findings of tumor mass growth, this growth being more subtle when CTC numbers varied little. Additionally, after ablative therapy with radiation, an increased number of CTCs was found in circulation, and the authors suggested as an explanation the prompt effusion of these cells after the therapeutic intervention [64].

## 8. Future Directions

It is unquestionable that CTCs have significant advantages as potential biomarkers for RCC. In addition to being obtained non-invasively, they reflect tumor heterogeneity, are exceedingly specific for RCC and allow for cell analysis at structural (based on physical properties and surface markers), molecular (through nucleic acids and proteins) and genomic (through sequencing) levels [105,106]. These features entail the use of CTCs in clinical practice, mostly in prognosis, prediction of response to treatment and disease monitoring. CTCs enumeration remains a key point in their use, requiring adequate and reliable methods for detection and isolation. Nonetheless, difficulties persist in CTC isolation, enrichment and detection. Therefore, it is mandatory to optimize the technologies currently used, increasing sensitivity and specificity, so that no CTC subtype falls outside the detection range. Furthermore, standardization, automation and cost reduction of existing technologies is a major concern, as they seem to be main contributors to the limited use of CTCs at present. Thus, in addition to technical sensitivity and specificity, simplicity and low operation cost are key to transferring CTCs’ characterization to the clinical setting. Eventually, this transition may be fostered by the adoption of combined strategies for CTC enrichment and detection based on their specific advantages.

Furthermore, and in addition to the improvement of technologies for CTCs’ isolation and collection, there is a clear need to invest in the development of multimodal platforms which may ally effective CTC capture with detailed characterization. Literature dealing with this topic remains scarce, eventually illustrating the difficulties of such a challenge. In our vision of the future, functional characterization of RCC-CTCs should include multi-omic analysis, i.e., at the genomic, proteomic, transcriptomic, epigenomic and metabolomic level. This will very likely provide invaluable information about RCC biology, enabling the identification of novel molecular targets for therapy and fostering the development of new drugs. Moreover, CTC characterization at the individual level may highlight unique aspects inherent to RCC, allowing for a more personalized approach.

Innovation in RCC research has been a constant in recent years, focusing on understanding the tumor microenvironment and deciphering the molecular mechanisms underlying pathogenesis that may aid in the development of new therapies. Besides the investigation of CTCs in liquid biopsy, their functional characterization using in vivo models may allow for a leap forward concerning drug resistance. In this context, dissemination of such a strategy of analysis would require models which might be easily available, with low cost and low impact on animal welfare. The chicken chorioallantoic membrane (CAM) assay meets these requirements, being used already in the investigation of CTCs’ role in metastatic progression [107]. This has constituted a significant advance in characterization of metastatic lung, breast and prostate cancer and may be applied to RCC soon.

From another broad perspective, the effectiveness and usefulness of CTCs as biomarkers may be expanded through their combination with other established biomarkers, also available through liquid biopsy, as already accomplished in colorectal cancer [108]. A seductive strategy comprises integration of CTCs, exosomes and cell-free DNA, combining the best of the three worlds, thus acquiring a greater knowledge on cancer biology. This is another line of reasoning that is now emerging and that, in the future, may have an application in RCC.

## 9. Conclusions

Renal cell carcinoma remains a significant clinical challenge, with substantial genotypic and phenotypic diversity, which determines the unpredictability of the disease course and challenging therapy selection. In this scenario, CTCs constitute an appealing biomarker, intimately related with the disease’s biology and behavior, enabling several potential clinical applications. At present, Cell Search™ remains the only FDA-approved technology, having several shortcomings. CTCs enumeration, which remains the only approved clinical use in a restricted number of tumor models, needs to be complemented by in-depth characterization at multi-omics and functional level to become an attractive routine tool, impacting disease management and fostering the use of new therapies in RCC. Several gaps need to be filled before CTCs may be more broadly used in clinical practice, which will require additional research and technological development.

## Figures and Tables

**Figure 1 cancers-15-00287-f001:**
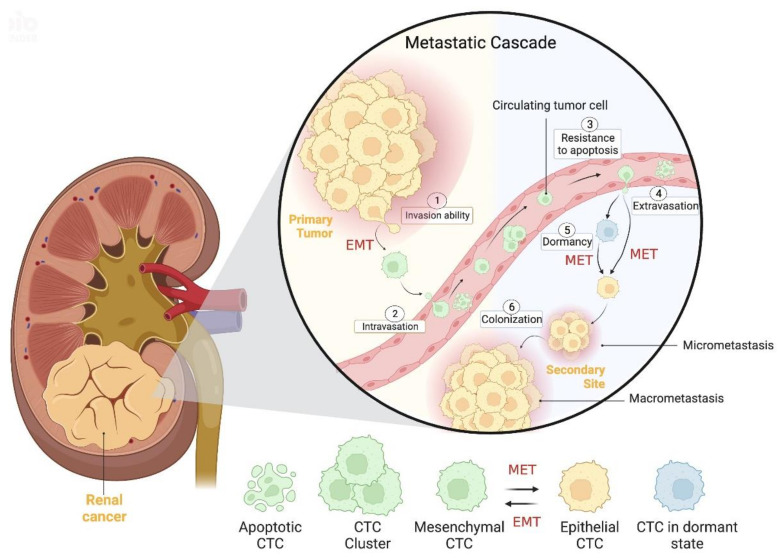
Schematic illustration of the metastatic cascade process in renal cell carcinoma and the role of circulating tumor cells (CTC). Created with BioRender.com.

**Figure 2 cancers-15-00287-f002:**
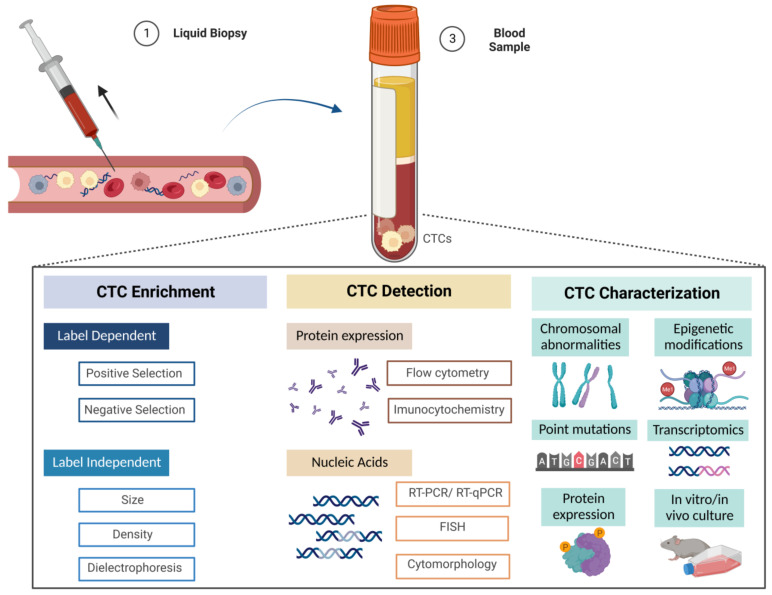
Schematic diagram of the steps involved in RCC-CTCs processing after blood collection: firstly, there is the step of CTC enrichment that can be achieved through various techniques; secondly, the detection step may be based on protein or nucleic acid expression and third, the characterization step may be accomplished based on several peculiarities of CTCs (transcriptomic, epigenetic, protein, mutational and cell culture). Created with BioRender.com.

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
