# Peer review of "Circulating Tumor Cells as Biomarkers for Renal Cell Carcinoma: Ready for Prime Time?"

_cancers, 2022, doi:10.3390/cancers15010287_

Round 1
Reviewer 1 Report
Very comprehensive and informative review - tabulated summaries would be particularly useful for a review of this type - see my detailed comments below:
Introductory sentence could be improved ..i.e. ‘14th position in the ranking'
Line 56 - sentence needs improvement
The use of the term entity for different subtypes of RCC seems odd - I realised it is used, but is it the best descriptor of different genetic subtypes?
Line 65 - reword the sentence without the second, third place reference
Line 79 - SPECT doesn’t seem defined
The referencing throughout needs to be edited - sometimes there is spacing left between the reference and text, sometime there is not. Position relative to full-stop is also inconsistent
Line 148 needs revision ...’finally, eventually’...
Line 151 - remove Nel et all showed
Line 173 is unclear - ' only a single of these CTCS? full stop before and after reference
Line 350 – ‘advantage is disclosed’ is not well written
The disadvantage of the Cell Search system - i.e. missing EpCam negative CTCs applies to all methodologies relying on EpCam staining - but this wasn’t highlighted earlier, and should not be highlighted as a CellSearch specific limitation
Line 378 - values inferior needs correcting
The NanoVelcro platform section indicates that they compared the new markers to EpCam only - and the details of this comparison should be expanded on. That is, how many matched patients had CTCs detected with these selection strategies
The authors mention lower purity rate with negative selection - but some specific details are important - what are the purity rates? A table summarising similar approaches with their advantages and disadvantages and requirements would be useful
The section 5.2 isn’t expanded on - or does 5.3 belong in this section?
Line 430 is not well written - 'disclose high variability?'
Line 447 - high volumes of blood mentioned - and it is important to highlight the differences - how much blood is required and how does this compare to other methods
Has low pass next generation genome sequencing been used for RCC CTC analysis?
Some discussion on the relative value of CTCs vs other liquid biopsies, in particular, ctDNA would be valuable
Author Response
Reviewer #1
Very comprehensive and informative review - tabulated summaries would be particularly useful for a review of this type - see my detailed comments below:
Re: We thank the positive comments of the Reviewer
Introductory sentence could be improved ..i.e. ‘14th position in the ranking'
Re: Thank you for the suggestion. The sentence was improved.
Line 56 - sentence needs improvement
Re: Thank you for the suggestion. The sentence was improved.
The use of the term entity for different subtypes of RCC seems odd - I realised it is used, but is it the best descriptor of different genetic subtypes?
Re: Thank you for the comment. “Entity” is the term employed in WHO classification. Considering that the authors of the classification are considered the paramount experts in the field, we believe that this is the best descriptor.
Line 65 - reword the sentence without the second, third place reference
Re: Thank you for the suggestion. The sentence was improved.
Line 79 - SPECT doesn’t seem defined
Re: Thank you for calling our attention to this. The definition was added, as well as that for PET/CT, in the line above.
The referencing throughout needs to be edited - sometimes there is spacing left between the reference and text, sometime there is not. Position relative to full-stop is also inconsistent
Re: Thank you for calling our attention to this. We thoroughly reviewed and corrected.
Line 148 needs revision ...’finally, eventually’...
Re: Thank you for the suggestion. The sentence was improved.
Line 151 - remove Nel et all showed
Re: Thank you for the suggestion. This was removed.
Line 173 is unclear - ' only a single of these CTCS? full stop before and after reference
Re: Thank you for the suggestion. The sentence was improved and corrected.
Line 350 – ‘advantage is disclosed’ is not well written
Re: Thank you for the suggestion. The sentence was improved.
The disadvantage of the Cell Search system - i.e. missing EpCam negative CTCs applies to all methodologies relying on EpCam staining - but this wasn’t highlighted earlier, and should not be highlighted as a CellSearch specific limitation
Re: Thank you for the suggestion. The sentence was modified to include that important information.
Line 378 - values inferior needs correcting
Re: Thank you for the suggestion. The sentence was corrected.
The NanoVelcro platform section indicates that they compared the new markers to EpCam only - and the details of this comparison should be expanded on. That is, how many matched patients had CTCs detected with these selection strategies
Re: Thank you for the suggestion. The information was added.
The authors mention lower purity rate with negative selection - but some specific details are important - what are the purity rates? A table summarising similar approaches with their advantages and disadvantages and requirements would be useful
Re: Thank you for your question. After a thorough investigation into the purity rates mentioned in the many articles we used in our review, none of them (including the ones that they cited) specify the particular details inherent to low purity.
The section 5.2 isn’t expanded on - or does 5.3 belong in this section?
Re: Thank you for calling our attention to this. Indeed, there is a mistake in numbering the subsections. This was corrected.
Line 430 is not well written - 'disclose high variability?'
Re: Thank you for the suggestion. The sentence was corrected.
Line 447 - high volumes of blood mentioned - and it is important to highlight the differences - how much blood is required and how does this compare to other methods
Re: Thank you for your suggestion. The information was added as well as a new reference (67) to support.
Has low pass next generation genome sequencing been used for RCC CTC analysis?
Re: Thank you for your question. Indeed, next generation genome sequencing has been used to analyse RCC-CTCs in recent years. For example, in reference 34 this strategy was used to analyse copy number alterations in RCC-CTCs.
Some discussion on the relative value of CTCs vs other liquid biopsies, in particular, ctDNA would be valuable
Re: Thank you for your suggestion. The information was added (paragraph starting in line 843), along with a new reference (100).
Reviewer 2 Report
In this review the Authors clearly and exhaustively describe the technical aspects and the potential diagnostic and prognostic role of CTCs in renal cell carcinoma. I only recommend a thorough revision of the text to eliminate some typos.
for example
See line 432; revise “CTCS”
See fig. 2; revise “imunocytochemistry”
make uniform the acronyms RCC-CTCs or RCC CTCs
Author Response
Reviewer #2
In this review the Authors clearly and exhaustively describe the technical aspects and the potential diagnostic and prognostic role of CTCs in renal cell carcinoma. I only recommend a thorough revision of the text to eliminate some typos.
for example
See line 432; revise “CTCS”
See fig. 2; revise “imunocytochemistry”
make uniform the acronyms RCC-CTCs or RCC CTCs
Re: We thank the positive comments of the Reviewer. A thorough revision was performed to correct the typos and uniformize the acronyms.
Round 2
Reviewer 2 Report
The review of the paper by the Authors is adequate. Thanks.